# Marine Bacterioplankton Community Dynamics and Potentially Pathogenic Bacteria in Seawater around Jeju Island, South Korea, via Metabarcoding

**DOI:** 10.3390/ijms241713561

**Published:** 2023-09-01

**Authors:** Hyun-Jung Kim, Kang Eun Kim, Yu Jin Kim, Hangoo Kang, Ji Woo Shin, Soohyun Kim, Sang Heon Lee, Seung Won Jung, Taek-Kyun Lee

**Affiliations:** 1Library of Marine Samples, Korea Institute of Ocean Science & Technology, Geoje 53201, Republic of Korea; hjkim8845@kiost.ac.kr (H.-J.K.); rkddmssl@kiost.ac.kr (K.E.K.); rladbwls06069@kiost.ac.kr (Y.J.K.); sjw3003@kiost.ac.kr (J.W.S.); soohyunk1122@kiost.ac.kr (S.K.); 2Department of Oceanography and Marine Research Institute, Pusan National University, Busan 46241, Republic of Korea; sanglee@pusan.ac.kr; 3Department of Ocean Science, University of Science & Technology, Daejeon 34113, Republic of Korea; 4Vessel Operation & Observation Team, Korea Institute of Ocean Science & Technology, Geoje 53201, Republic of Korea; hgkang@kiost.ac.kr; 5Ecological Risk Research Department, Korea Institute of Ocean Science & Technology, Geoje 53201, Republic of Korea

**Keywords:** metabarcoding, bacterioplankton community, 16S rRNA, *Candidatus Pelagibacter*, *Pseudoalteromonas*, *Vibrio*, pathogenic bacteria

## Abstract

Understanding marine bacterioplankton composition and distribution is necessary for improving predictions of ecosystem responses to environmental change. Here, we used 16S rRNA metabarcoding to investigate marine bacterioplankton diversity and identify potential pathogenic bacteria in seawater samples collected in March, May, September, and December 2013 from two sites near Jeju Island, South Korea. We identified 1343 operational taxonomic units (OTUs) and observed that community diversity varied between months. *Alpha-* and *Gamma-proteobacteria* were the most abundant classes, and in all months, the predominant genera were *Candidatus Pelagibacter*, *Leisingera*, and *Citromicrobium*. The highest number of OTUs was observed in September, and *Vibrio* (7.80%), *Pseudoalteromonas* (6.53%), and *Citromicrobium* (6.16%) showed higher relative abundances or were detected only in this month. Water temperature and salinity significantly affected bacterial distribution, and these conditions, characteristic of September, were adverse for *Aestuariibacter* but favored *Citromicrobium*. Potentially pathogenic bacteria, among which *Vibrio* (28 OTUs) and *Pseudoalteromonas* (six OTUs) were the most abundant in September, were detected in 49 OTUs, and their abundances were significantly correlated with water temperature, increasing rapidly in September, the warmest month. These findings suggest that monthly temperature and salinity variations affect marine bacterioplankton diversity and potential pathogen abundance.

## 1. Introduction

Marine bacterioplankton have essential ecosystem functions owing to their ecological connections with other organisms, and their distribution and abundance in marine environments are affected by several environmental and geographic factors [1,2]. Understanding how these environmental factors affect their composition and distribution may help to elucidate their relationships with other organisms and improve predictions of ecosystem responses to environmental changes [3]. In particular, examining the effects of seasonal variations in environmental and geographic factors on community dynamics is essential to elucidate their ecological roles [4,5,6].

Temporal and spatial changes in bacterioplankton communities in response to environmental change have been examined in recent studies [7,8]. Marine bacterioplankton communities, which are finely structured, exhibit discrete phylogenetic clustering and abundant diversity [9]. Metabarcoding, which characterizes taxonomic diversity and community structure based on the analysis of 16S rRNA gene sequences, has been used to reveal the high diversity of marine bacterioplankton [10,11,12]. We previously applied metabarcoding to investigate bacterioplankton communities and species composition, monthly changes, and potential pathogenic bacteria [13,14,15,16,17,18]. For instance, in Goseong Bay (South Korea), *Bacteroidetes* and *Actinobacteria* were identified as the most abundant phyla [19]. Studies on year-round bacterioplankton diversity may help in forecasting seasonal changes [6], and metabarcoding may provide insights into the intricate relationships within bacterioplankton communities owing to changes in various environmental factors [5].

To elucidate these factors, we used metabarcoding to investigate marine bacterioplankton community dynamics and identify potentially pathogenic bacteria in seawater around Jeju Island, South Korea, in four months in 2013. Our findings elucidated the distribution of marine bacterioplankton in the waters around this site and established a baseline for further studies. These findings will improve the prediction of marine bacterioplankton responses to changes in environmental factors.

## 2. Results and Discussion

### 2.1. Seawater Environmental Characteristics

We sampled seawater around Jeju Island, South Korea, in March, May, September, and December 2013 to characterize the bacterioplankton community in this area, considering seasonal changes in seawater temperature and salinity. Water temperature (range 17.0–25.3 °C) varied seasonally, consistent with the patterns commonly observed for temperate environments: highest in summer and lowest in winter [20,21]. Salinity ranged from 18.5 to 34.4 and was lowest in September. In South Korea, the summer–autumn rainy season corresponds to the period from mid-August to mid-September, when typhoons affect the Korean Peninsula directly or indirectly [22]. The large input of rainwater during this period rapidly reduces seawater salinity, and this affects the microbial community in this area [23,24].

### 2.2. Metabarcoding Results

#### 2.2.1. OTU Diversity

Following quality control filtering, we obtained 8003 reads (average number of reads per sample) (Table 1). These reads corresponded to 1343 bacterioplankton operational taxonomic units (OTUs). Ranking by month showed the highest number of OTUs in September, followed by May, December, and March (Figure 1). The high taxonomic diversity and richness in September may be related to the low salinity and high temperature of the seawater within this month. These findings are consistent with those of other studies, which revealed that bacterial species diversity is particularly high in September owing to these climate conditions [25,26]. In autumn, interspecific interactions between bacterioplankton weaken, resulting in lower diversity [24]. Further, the active mixing of the water column during this period alters the water column’s physical and chemical parameters; this, in turn, leads to significant seasonal differences in diversity and composition [27].

#### 2.2.2. Diversity of Bacterioplankton

The identified bacterioplankton OTUs were assigned to 21 phyla. The most abundant phyla, accounting for 98.25% of the total relative abundance, were *Proteobacteria*, *Bacteroidetes*, and *Cyanobacteria* (Figure 2a). The relative abundance of *Proteobacteria* was slightly lower in May (at 78.86%) but was >85% in the other months. The second most abundant phylum was *Bacteroidetes*. Its abundance increased from 12.54% in March to 19.23% in May and declined to 6.39% in September. Further, *Cyanobacteria* were two to five times more abundant in September (3.62%) than in the other months.

Class-level diversity varied considerably between months (Figure 2b). Specifically, 47 classes were identified, with *Alphaproteobacteria*, *Gammaproteobacteria*, *Flavobacteriia*, and *Cyanophyceae* showing dominance in all of the months. *Alphaproteobacteria* was more abundant in September (61.84%) and December (62.83%) than in May (35.58%) and March (53.61%). *Gammaproteobacteria* and *Flavobacteriia* were more abundant in May (42.66% and 19.03%, respectively) than in the other months. *Cyanophyceae* exhibited 1.48% relative abundance in May and was more abundant in September (3.59%). However, in December, its abundance decreased (1.25%).

#### 2.2.3. Changes in Common Bacterioplankton

The genus-level heat map of the bacterioplankton community revealed distinct changes in diversity over the months (Figure 3a). Of the 719 genera identified, 12 accounted for >5% of the total bacterioplankton relative abundance. Via nonmetric multidimensional scaling (NMDS) analysis, the bacterioplankton community was classified into two groups, considering a 35% similarity level (Figure 3b). Thus, *Candidatus Pelagibacter* (*Alphaproteobacteria*) was identified as the most abundant genus in all the months and was most abundant in December (29.24%). *Aestuariibacter* (*Gammaproteobacteria*), the second most abundant genus, was detected in all months except September and was most abundant in March (23.18%). Further, the highest abundances of *Lentibacter* (8.33%), *Nereida* (7.20%), *Loktanella* (5.54%), and *Aliiroseovarius* (5.44%), belonging to *Alphaproteobacteria*, were recorded in March, and the highest abundances of *Marinomonas* (11.17%; *Gammaproteobacteria*) and *Aurantivirga* (10.16%; *Bacteroidetes*) were recorded in May. September showed the highest abundances of *Vibrio* (7.80%) and *Pseudoalteromonas* (6.53%), which are *Gammaproteobacteria*. Furthermore, *Citromicrobium* (6.16%; *Alphaproteobacteria*) was detected only in September, and December showed the highest abundance of *Leisingera* (22.0%; *Alphaproteobacteria*).

*Candidatus Pelagibacter ubique*, which accounts for most species of the genus *Pelagibacter*, is distributed in marine environments at temperatures in the range of 12–15 °C and can be cultured at 18 °C [28,29]. In this study area, the water is slightly warmer in September, but still within this range; however, salinity varies substantially, in the range of 18–34. Reportedly, *Candidatus Pelagibacter ubique* grows under oligohaline–mesohaline conditions at salinity values ≥ 30 [29,30]. Thus, the temperature and salinity conditions of the seawater at our study site favor the growth of species of the genus *Pelagibacter*.

For *Aestuariibacter halophilus*, the most abundant species of this genus, the optimum temperature and pH for growth are 40 °C and pH 7–8, respectively, but its growth is still possible at 15–40 °C. This species exhibits strict halophilicity; hence, it requires the saline conditions of seawater for growth [31]. Low salinity, therefore, limited its growth in this region in the month of September. *Citromicrobium*, detected only in September, comprised one species, *C. bathyomarinum*, which grows under temperature, salinity, and pH conditions in the ranges 20–42 °C, 0–10, and 6.0–8.0, respectively [32]. Thus, the water around Jeju Island is most suitable for *C. bathyomarinum* in September. Given that bacterial growth is affected by various environmental factors, it is difficult to clarify the effects of water temperature and salinity in this regard. In this study, however, these two factors were identified as the most important factors affecting bacterial growth and distribution.

We performed LEfSe (Linear discriminant analysis Effect Size) analyses to identify the abundant bacterioplankton taxon based on a comparison between September and the other months (March, May, and December) (Figure 4; Appendix A). Based on the results obtained, *Cyanophyceae*, *Alphaproteobacteria* (*Kiloniellales*, *Rhodospirillales*, *Sphingomonadales*), *Saprospirales*, and *Pseudomonadales* were identified as the major bacterioplankton in September. Conversely, *Bacteroidetes* and *Gammaproteobacteria* (*Pseudomonadales*, *Alteromonadales*, and *unclassified Gammaproteobacteria*) constituted the common taxa in Jeju seawaters in the other months.

### 2.3. Identification of Potential Pathogens

At the class level, we identified the following potentially pathogenic bacteria: *Gammaproteobacteria* (44 OTUs), *Flavobacteriia* (three OTUs), *Betaproteobacteria* (one OTU), and *Epsilonproteobacteria* (one OTU) (Appendix A). In particular, *Vibrio* (28 OTUs) and *Pseudoalteromonas* (six OTUs) were the most abundant potentially pathogenic genera (Figure 5, Appendix A), and were most abundant in September. Further, potential pathogenic bacteria showed significantly increased relative abundances in September (*p* < 0.05, one-way ANOVA).

The relative abundance of potentially pathogenic bacteria was significantly and positively correlated with water temperature (Spearman correlation analysis; *p* < 0.05). In particular, the relative abundance of *Pseudoalteromonas* showed a highly significant correlation with water temperature (*p* < 0.01, ρ = 0.835). For *Pseudoalteromonas*, the optimal temperature for growth is >20 °C [33]. Further, *Vibrio* spp. were 3.5–14 times more abundant in September than in the other months. Reportedly, *Vibrio* grows at temperatures > 20 °C (similar to the temperature of the seawater around Jeju Island in September) and can tolerate salinity in the range 3–37 [34]. It has also been reported that *Vibrio* abundance is strongly associated with water temperature [35,36]. *Vibrio* and *Pseudoalteromonas* were the most common potential pathogens detected in the waters of Kaneohe Bay, Hawaii [37]. These potentially pathogenic bacteria can infect various marine plants and animals, including humans (Appendix A). In particular, *V. kanaloae*, *V. tasmaniensis*, *V. chagasii*, *P. tetraodonis*, and *P. nigrifaciens* infect marine animals, such as lobsters, fish, oysters, and sea cucumbers.

## 3. Materials and Methods

### 3.1. Sample Collection

Seawater samples (1 L) were collected from a 1 m depth at two sites around Jeju Island (Site 1, 33°22′89″ N, 126°56′53″ E; Site 2, 33°23′53″ N, 126°56′53″ E) on 10 March, 21 May, 8 September, and 10 December 2013 (Figure 6). The seawater temperature and salinity at these sampling points were obtained from the Marine Environment Information System of Korea (http://www.meis.go.kr, accessed on 15 May 2014). Thereafter, 1 L samples of the collected seawater were each passed through a polycarbonate filter (3 µm pore size) (TSTP04700, Millipore, Ireland), harvested, and further passed through a 0.2 µm filter (A020A047A, Advantec MFS Inc., Tokyo, Japan). Next, they were stored at 4 °C until genomic DNA extraction.

### 3.2. Metabarcoding Analysis

The filters containing the microbes were cut into eight pieces before DNA extraction. Thereafter, total microbial DNA was isolated using the PowerSoil DNA Isolation Kit (MoBio, Solana Beach, CA, USA) according to the manufacturer’s instructions. The extracted genomic DNA was then subjected to PCR amplification using primers targeting the V1–V3 region of the 16S rRNA gene [18], i.e., 27F forward primer, 5′-GAG TTT GAT CMT GGC TCA G-3′ and 518R reverse primer, 5′-ATT ACC GCG GCT GCT GG-3′. Partial 16S rDNA gene sequences were used to analyze bacterial diversity. Given that the divergence level varies between the regions of the 16S rDNA gene, the choice of partial sequence regions can significantly affect analysis results [38]. Thus, it was important to determine whether a partial 16S rDNA sequence region could support bacterial characterization as reliably as nearly full-length 16S rDNA genes.

Each primer was tagged using multiplex identifier (MID) adaptors (Roche, Mannheim, Germany) following the manufacturer’s instructions. The use of MID adaptors enabled the automatic sorting of metabarcoding-derived sequencing reads. Further, amplification was performed under the following conditions: pre-denaturation at 95 °C for 5 min; 30 cycles of denaturation at 95 °C for 30 s, primer annealing at 55 °C for 30 s, elongation at 72 °C for 30 s; and final elongation at 72 °C for 5 min. The PCR products were purified using a QIAquick PCR Purification Kit (cat. 28106; Qiagen, Hilden, Germany). Similar amounts of extracted PCR products were pooled, and short fragments (non-target products) were eliminated using the AMPure Bead Kit (Agencourt Bioscience, Beverly, MA, USA). PCR product size and quality were further evaluated using a Bioanalyzer 2100 device (Agilent, Palo Alto, Foster City, CA, USA). Thereafter, sequencing was conducted using the 454 GS Junior Sequencing System (Roche Applied Science, Penzberg, Germany) following the manufacturer’s instructions.

### 3.3. Bioinformatics Analysis

Following metabarcoding, bioinformatics analysis was performed as previously described [18]. After sequencing, the quality check was performed to remove short sequence reads (<150 bp), low-quality sequences (quality score < 25), singletons, chloroplast sequences, non-bacterial ribosomal sequences, and chimeras [39,40]. Then, using the Basic Local Alignment Search Tool (BLAST v. 2.14.0), the sequence reads obtained were compared to the sequences in the Silva rRNA database. Similar sequence reads (E-value < 0.0001) were considered partial 16S rDNA sequences, and the taxonomic level (class or genus) of the most similar sequence in the rRNA database was assigned to each of the identified sequence reads. To analyze OTUs, CD-HIT-OTU software was used for clustering [41], while Mothur platform (v 1.35.1) was used to estimate Shannon–Weaver diversity and Chao1 richness [42]. The taxonomy of the sequence with the highest similarity was assigned to the sequence read (species or genus levels with >98 or >94%, respectively).

### 3.4. Statistical Analysis and Selection of Potentially Pathogenic Bacteria

Data were presented as the mean of samples from two sampling sites. To compare bacterioplankton community abundances among the four months, hierarchical clustering analysis was performed via “group average” clustering using the Bray–Curtis dissimilarity method. This generated a ranked similarity matrix in which the rows represented the rankings of the column cases based on their similarity to the corresponding row case. OTU class and family relative abundances were square-root normalized for comparison. Normal distributions were assessed using the Kolmogorov–Smirnov test. We also considered OTUs with a relative abundance > 1% in at least one sample as ‘abundant’ and pooled the remainder.

To examine the relationships between measured parameters, Spearman correlation analysis was employed. An ordination plot was produced via nonmetric multidimensional scaling (NMDS) using a ranked similarity matrix. Further, the clustering, NMDS, and correlation analyses were performed using PRIMER 6 (v 6.1.13). Alpha diversity metrics and OTUs were analyzed using the ‘vegan’ package [43] in R Studio (v. 1.3.959). Heat maps of the relative abundances of the most abundant OTUs were plotted using ggplot2 [44] in R Studio. The investigated months were compared via one-way analysis of variance (ANOVA), followed by Scheffe’s post hoc test. *p* values < 0.05 were considered statistically significant.

To perform linear discriminant analysis (LDA), the Kruskal–Wallis test was employed to assess differences among classes (α < 0.05), while the pairwise Wilcoxon test, acting as a non-parametric analog, was conducted to compare subclasses (α < 0.05) [45,46]. The threshold on the logarithmic LDA score was set to 2.0 to determine discriminative features [47,48]. After the analysis was completed, the number of differentially abundant OTUs identified using each tool was assessed at an α level of 0.05. Finally, the potential presence of taxonomic groups that may explain the difference between bacterioplankton communities in different samples was explored using Linear discriminant analysis Effect Size (LEfSe, v 1.0) in the Galaxy framework.

The selection of potentially pathogenic bacteria was based on the literature. The characteristics and references on the basis of which these potential pathogenic bacteria were chosen are shown in Appendix A.

## 4. Conclusions

In this study, we examined the variation in bacterioplankton diversity, community composition, and the presence of potentially pathogenic bacteria in the waters around Jeju Island for four months in 2013. Thus, we observed that community diversity was highest in September, within which the highest abundance of potentially pathogenic bacteria was also observed. This is the first study describing species-level diversity and several pathogenic bacteria in seawater in the study region. These findings regarding potentially pathogenic bacteria can be used to develop an early warning system with respect to marine pathogens.

## Figures and Tables

**Figure 1 ijms-24-13561-f001:**
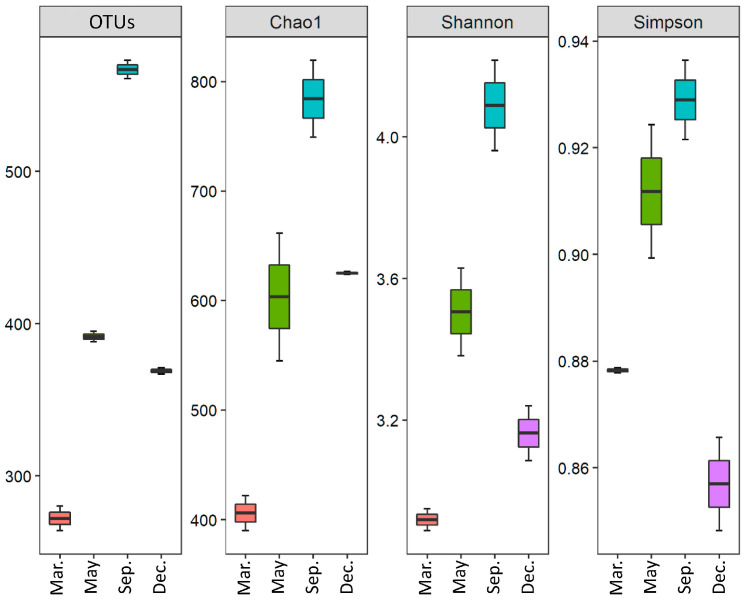
Variation of the number of sequences, operational taxonomic units, and diversity indices (Chao1, Shannon, and Simpson indices) for the marine bacterioplankton community in seawater around Jeju Island, South Korea, in four months in 2013.

**Figure 2 ijms-24-13561-f002:**
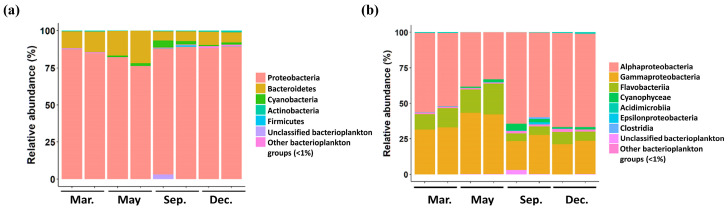
Monthly variation of bacterioplankton community relative abundance in seawater around Jeju Island, South Korea, in four months in 2013. (**a**) Phylum and (**b**) class level. Taxa with relative abundance < 1.0% were categorized as other bacterioplankton groups.

**Figure 3 ijms-24-13561-f003:**
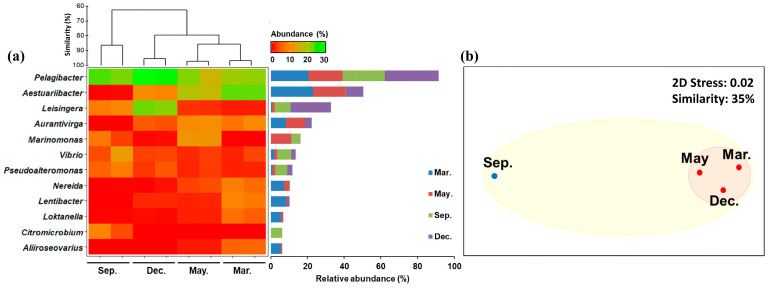
Heatmap of the relative abundances of the operational taxonomic units (OTUs) of common bacterioplankton (for OTUs with relative abundance > 1% in at least one sample) in seawater samples collected around Jeju Island, South Korea, in four different months in 2013. (**a**) Hierarchical agglomerative clustering results based on the group average of the relative abundances of the bacterioplankton OTUs. (**b**) Nonmetric multidimensional scaling (NMDS) plot for the bacterioplankton community based on the Bray–Curtis dissimilarity method. The relative abundances were square-root normalized.

**Figure 4 ijms-24-13561-f004:**
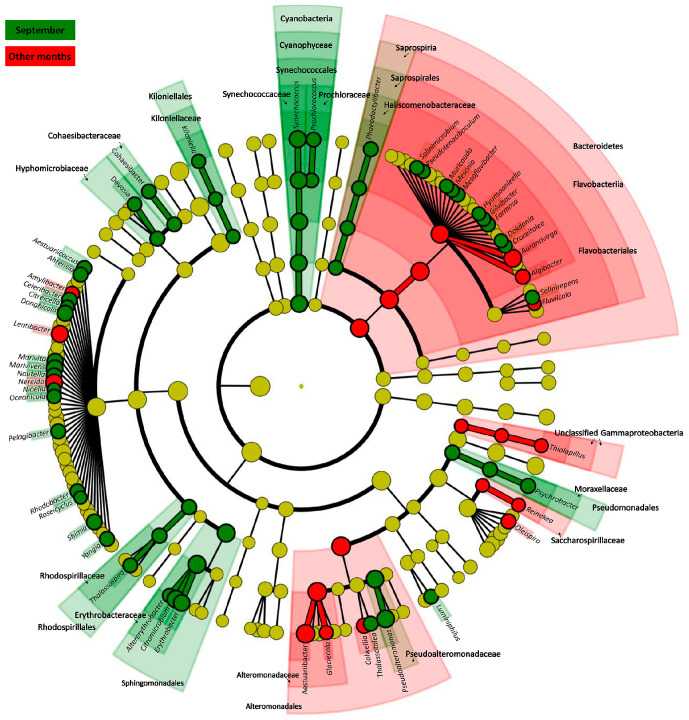
Cladograms of attached bacterioplankton lineages differ significantly between September and other months. The associated bacterioplankton groups at phylum to genus levels are listed from the center to the outside. The circle diameters are proportional to bacterioplankton taxon abundance. Significant discriminatory nodes are colored, and branch areas are shaded according to the highest-ranked group for the given taxon. Green and red areas indicate September and other months, respectively.

**Figure 5 ijms-24-13561-f005:**
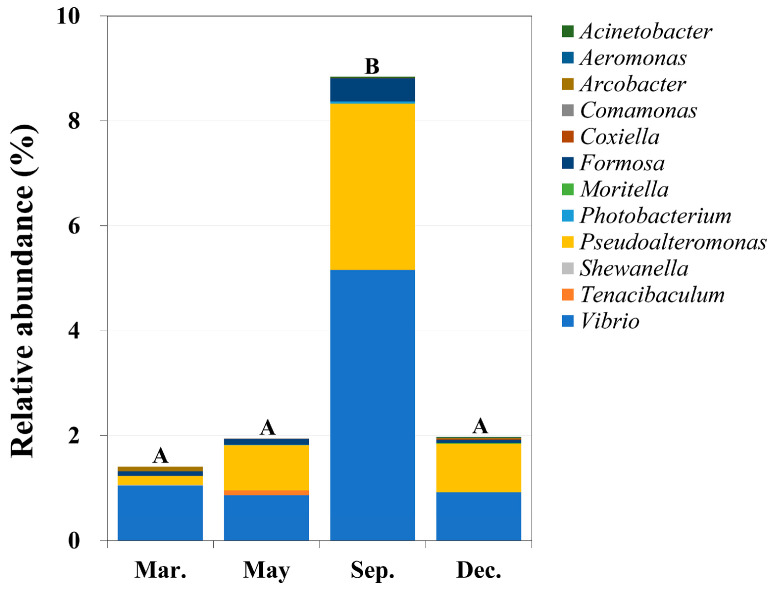
Genus-level relative abundances of potentially pathogenic bacteria in seawater around Jeju Island, South Korea, in four months in 2013. For each genus listed, >1% of the species in the genus are potentially pathogenic. These results were obtained by performing one-way ANOVA and Scheffe’s post hoc test. The letters A and B indicate significant differences among months (*p* < 0.05).

**Figure 6 ijms-24-13561-f006:**
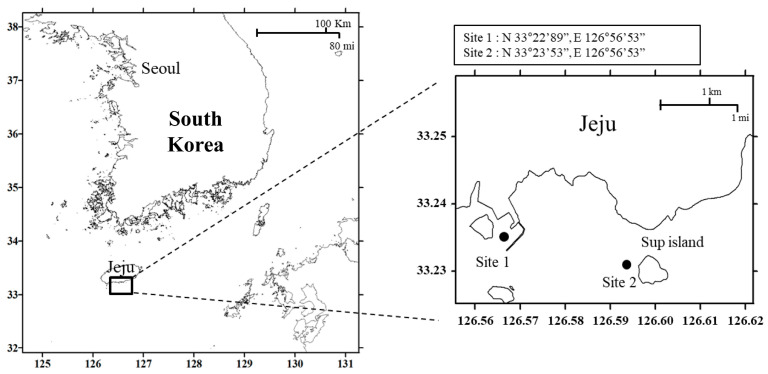
Seawater sampling sites around Jeju Island, South Korea, in four different months in 2013.

**Table 1 ijms-24-13561-t001:** Mean read counts, observed operational taxonomic units (OTUs), and alpha-diversity (Chao1, Shannon, and Simpson indices) for marine bacterioplankton in seawater around Jeju Island, South Korea, in four months in 2013.

Month	Trimmed Reads	OTUs	Chao1	Shannon	Simpson
March	6673	272	406.1	2.92	0.88
May	9782	393	603.4	3.51	0.91
September	9813	567	784.2	4.09	0.93
December	5745	369	625.1	3.16	0.86

## Data Availability

The datasets presented in this study can be found in online repositories. The repository and accession number PRJNA999943 (https://www.ncbi.nlm.nih.gov/genbank, accessed on 1 August 2023).

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
