# Peer review of "Marine Bacterioplankton Community Dynamics and Potentially Pathogenic Bacteria in Seawater around Jeju Island, South Korea, via Metabarcoding"

_ijms, 2023, doi:10.3390/ijms241713561_

Round 1
Reviewer 1 Report
The samples were collected in 2013, ten years ago. When it compared with recent samples, it would be meaningful. I do not think the meaningful of this type of study just based on ten years old samples.
1. The sequencing reads of each sample was too small to get reasonable results.
2. the total number of OTUs should not be the sum of 4 samples. All samples should be analyzed together. And the ASVs level analysis would be better than OTUs. It could be analyzed by commonly used software, such as QIIME 2.
3. The method of pathogenic bacteria detection should be more clearly.
NO
Author Response
Reviewer 1.
- The samples were collected in 2013, ten years ago. When it compared with recent samples, it would be meaningful. I do not think the meaningful of this type of study just based on ten years old samples.
Response
We thank the Reviewer for this valuable comment. Currently, we are conducting a study on the distribution and function of marine bacterioplankton in various sea areas via 16s V4–V5 metabarcoding using the Illumina Mi-Seq platform, and are submitting manuscripts for publication in this regard (1-6). The results of this study were obtained via analysis using Roche's GS Junior Sequencing System. Thus, accurately comparing these findings with those obtained using the different analysis method and a variable gene region is challenging. Currently, samples collected from the poles, such as those from the Antarctic and Arctic zones, and those collected at the equator, and from various sea areas, such as the Korean coast, the Atlantic Ocean, and the Pacific Ocean, are being analyzed. Thus, we think that it is more reasonable to compare the present results with those of these studies that are underway.
Selected References
- Jung, S.W.; Noh, S.Y.; Kang, D.; Lee, T-K., Comparison of bacterioplankton communities between before and after inoculation with an algicidal material, Ca-aminoclay, to mitigate Cochlodinium polykrikoides blooms: assessment using microcosm experiments. Journal of Applied Phycology. 2017, 29, 1343-1354.
- Kang, J.; Park, J.S.; Jung, S.W.; Kim, H-J.; Joo H.M.; Kang D.; Seo, H.; Kim, S.; Jang, M.C.; Lee, K-W.; Oh, S.J.; Lee, S.; Lee, T-K., Zooming on dynamics of marine microbial communities in the phycosphere of Akashiwo sanguinea (Dinophyta) blooms. Molecular Ecology, 2021, 30, 207-221.
- Jung, S.W.; Kang, J.; Park, J.S.;Joo, H.M.; Suh, S.S.; Kang. D.; Lee, T-K.; Kim, H-J., Dynamic bacterial community response to Akashiwo sanguinea (Dinophyceae) bloom in indoor marine microcosms. Scientific Reports, 2021, 11, 6983.
- Kim, H-J.; Park, J.S.; Lee, T-K.; Kang, D.; Kang, J.H.; Shin, K.; Jung, S.W., Dynamics of marine bacterial biofouling communities after initial Alteromonas genovensis biofilm attachment to anti-fouling paint substrates. Marine Pollution Bulletin, 2021, 172, 112895.
- Park, J.S.; Han, J.; Suh, S.S.; Kim, H-J.; Lee, T-K.; Jung, S.W.; Characterization of bacterial community structure in two alcyonacean soft corals (Litophyton and Sinularia sp.) from Chuuk, Micronesia. Coral Reefs, 2022, 41, 563–574.
- Kim, H-J.; Kim, K.E.; Park, J.S.; Kang, D.; Baek, S.H.; Lee, C.Y.; Kim, H.; Cho, S.; Lee, T-K.; Jung, S.W., Co-variance between free-living bacteria and Cochlodinium polykrikoides (Dinophyta) harmful algal blooms, South Korea. Harmful Algae, 2022, 122, 102371.
- The sequencing reads of each sample was too small to get reasonable results.
Response
We are grateful to the Reviewer for this valuable comment. Given that OTUs represent the number (diversity) of bacteria, we have included the number of trimmed read counts to ensure a more accurate description. Please, see revised Table 1 in the manuscript, which is also shown below.
Table 1. Mean number of sequences, Reads, observed operational taxonomic units (OTUs), and alpha-diversity (Chao1, Shannon, and Simpson indices) for marine bacterioplankton in seawater around Jeju Island, South Korea, in four months in 2013.
|
Month |
Trimmed Reads |
OTUs |
Chao1 |
Shannon |
Simpson |
|
March |
7,483 |
272 |
406.1 |
2.92 |
0.88 |
|
May |
6,784 |
393 |
603.4 |
3.51 |
0.91 |
|
September |
9,473 |
567 |
784.2 |
4.09 |
0.93 |
|
December |
5,982 |
369 |
625.1 |
3.16 |
0.86 |
- the total number of OTUs should not be the sum of 4 samples. All samples should be analyzed together. And the ASVs level analysis would be better than OTUs. It could be analyzed by commonly used software, such as QIIME 2.
Response
We absolutely agree with Reviewer’s comment that ASV analysis is more accurate than OTU analysis. Presently, our results are predominantly based on ASV analysis performed via DADA2. However, the results of this study were interpreted based on the V1–V3 region using QIIME analysis without performing DADA2 analysis for the 16S V3–V4 region.
- The method of pathogenic bacteria detection should be more clearly.
Response
The selection of potentially pathogenic bacteria was based on the literature. The characteristics and references on the basis of which these potential pathogenic bacteria were chosen are shown in Table S2. Please, see Page 8, Lines 296–298 in the revised manuscript.

Reviewer 2 Report
There are a few notes to this good article.
1) Lines 240-242: Reference 38 does not mention the V1-V3 region and primers 27F and 518R.
2) What is the length of the resulting product?
3) What are the similarity thresholds for assigning sequences to taxa of different ranks?
4) From what depth were water samples taken to extract the bacteria? To which horizon do the values of temperature and salinity belong?
5) Ref [42] describes CD-HIT-OTU-MiSeq package for Illumina data. You should refer to the articleLi W, Fu L, Niu B, Wu S, Wooley J. Ultrafast clustering algorithms for metagenomic sequence analysis. Brief Bioinform. 2012;13(6):656-68. doi: 10.1093/bib/bbs035.
Author Response
Reviewer 2.
- Lines 240-242: Reference 38 does not mention the V1-V3 region and primers 27F and 518R.
Response
We thank the reviewer for the keenness applied in reviewing our manuscript. We have corrected the references for accuracy. Please, see Page 7, Lines 236–237 in the revised manuscript.
- What is the length of the resulting product?
Response
The length of V1–V3 region is approximately 500 bp, and Table 1 has been revised to explain the results in this regard in detail. Further, the revised table also includes the number of sequences and the number of reads produced. Please, see Page 2, Lines 92–95 in the revised manuscript.
Table 1. Mean number of sequences, Reads, observed operational taxonomic units (OTUs), and alpha-diversity (Chao1, Shannon, and Simpson indices) for marine bacterioplankton in seawater around Jeju Island, South Korea, in four months in 2013.
|
Month |
Trimmed Reads |
OTUs |
Chao1 |
Shannon |
Simpson |
|
March |
7,483 |
272 |
406.1 |
2.92 |
0.88 |
|
May |
6,784 |
393 |
603.4 |
3.51 |
0.91 |
|
September |
9,473 |
567 |
784.2 |
4.09 |
0.93 |
|
December |
5,982 |
369 |
625.1 |
3.16 |
0.86 |
- What are the similarity thresholds for assigning sequences to taxa of different ranks?
Response
We thank the Reviewer for this important question. In this study, we attempted to identify bacterial species with a genetic similarity of 98% or higher. However, when the species could not assigned at this 98% level, the similarity at the genus level defined at 94%. Please, see Page 8, Line 268-269.
- From what depth were water samples taken to extract the bacteria? To which horizon do the values of temperature and salinity belong?
Response
To measure bacteria distribution, we collected water samples at a depth of 1 m. Water temperature and salinity measurements were also performed at a depth of 1 m. We have added details in this regard in the revised manuscript. Please, see Page 7, Line 225.
- Ref [42] describes CD-HIT-OTU-MiSeq package for Illumina data. You should refer to the articleLi W, Fu L, Niu B, Wu S, Wooley J. Ultrafast clustering algorithms for metagenomic sequence analysis. Brief Bioinform. 2012;13(6):656-68. doi: 10.1093/bib/bbs035.
Response
We have checked the made corrections in accordance with the comment of the reviewer. Please, see Page 11, Lines 405-406 in the revised manuscript.

Round 2
Reviewer 1 Report
OK, no further comment.
OK, no further comment.